# Contribution of Drugs Interfering with Protein and Cell Wall Synthesis to the Persistence of *Pseudomonas aeruginosa* Biofilms: An In Vitro Model [note 1]

**DOI:** 10.3390/ijms22041628

**Published:** 2021-02-05

**Authors:** Gianmarco Mangiaterra, Elisa Carotti, Salvatore Vaiasicca, Nicholas Cedraro, Barbara Citterio, Anna La Teana, Francesca Biavasco

**Affiliations:** 1Department of Life and Environmental Sciences, Polytechnic University of Marche, via Brecce Bianche, 60131 Ancona, Italy; e.carotti@pm.univpm.it (E.C.); s.vaiasicca@staff.univpm.it (S.V.); n.cedraro@pm.univpm.it (N.C.); a.lateana@univpm.it (A.L.T.); f.biavasco@staff.univpm.it (F.B.); 2Department of Biomolecular Science, Biotechnology Section, University of Urbino “Carlo Bo”, via Arco d’Augusto 2, 61032 Fano, Italy; barbara.citterio@uniurb.it

**Keywords:** *Pseudomonas aeruginosa*, biofilm, persisters, viable but non culturable forms, flow cytometry, qPCR, antibiotic treatment, green fluorescent protein

## Abstract

The occurrence of *Pseudomonas aeruginosa* (PA) persisters, including viable but non-culturable (VBNC) forms, subpopulations of tolerant cells that can survive high antibiotic doses, is the main reason for PA lung infections failed eradication and recurrence in Cystic Fibrosis (CF) patients, subjected to life-long, cyclic antibiotic treatments. In this paper, we investigated the role of subinhibitory concentrations of different anti-pseudomonas antibiotics in the maintenance of persistent (including VBNC) PA cells in in vitro biofilms. Persisters were firstly selected by exposure to high doses of antibiotics and their abundance over time evaluated, using a combination of cultural, qPCR and flow cytometry assays. Two engineered GFP-producing PA strains were used. The obtained results demonstrated a major involvement of tobramycin and bacterial cell wall-targeting antibiotics in the resilience to starvation of VBNC forms, while the presence of ciprofloxacin and ceftazidime/avibactam lead to their complete loss. Moreover, a positive correlation between tobramycin exposure, biofilm production and c-di-GMP levels was observed. The presented data could allow a deeper understanding of bacterial population dynamics during the treatment of recurrent PA infections and provide a reliable evaluation of the real efficacy of the antibiotic treatments against the bacterial population within the CF lung.

## 1. Introduction

*Pseudomonas aeruginosa* lung infections represent the main cause of morbidity and mortality for Cystic Fibrosis (CF) patients, who suffer from recurrent pulmonary exacerbations and decreased lung function [1]. Patients are often subjected to heavy and cyclic antibiotic treatments [2], which fail to completely eradicate the pathogen though, since, after a time period of apparent infection clearance, the symptoms recrudescence is observed and the same *P. aeruginosa* strain is isolated again in sputum bacterial culture [3]. Such recurrence is attributed to the occurrence of persistent bacterial forms, which have been defined as cells able to tolerate high bactericidal antibiotic concentrations, without the involvement of a heritable genetic factor [4]. Moreover, some authors [5,6] reported the detection of viable but non-culturable (VBNC) cells in hospitalized CF patients after the first week of antibiotic treatment. VBNC cells are specialized persisters unable to grow on microbiological media but still live and capable of regaining culturability under specific conditions [7]. Both these phenotypes allow bacteria to survive unfavorable conditions, including antibiotic pressure, and to proliferate again once the stress factor is removed. Antibiotic treatment can either select pre-existing persisters (either stochastic or triggered by a variety of stress conditions, above all starvation [8]) or induce itself their development [4] as either culturable persisters or VBNC cells [9]. An interesting hypothesis is the involvement of low drug concentrations in the development and maintenance of these specific phenotypes [10,11], according to the hormesis theory, which describes the dual, concentration-dependent effect of antimicrobial compounds, i.e., bactericidal at high doses and gene expression regulators at subinhibitory concentrations [12].

We have long investigated the role of different antibiotics and antibiotic concentrations in the development of *Staphylococcus aureus* and *P. aeruginosa* VBNC cells [10,11,13]. Moreover, by using a species-specific qPCR protocol, we have demonstrated the presence of VBNC *P. aeruginosa* in sputum samples of intermittent and/or chronic CF patients [14], and have described the different involvement of subinhibitory tobramycin or ciprofloxacin concentrations in the maintenance of a *P. aeruginosa* VBNC subpopulation in an in vitro biofilm model, exposed to both starvation and low antibiotic concentrations. The use of flow cytometry, after live/dead staining, allowed us to confirm the viability of the non-culturable subpopulation [11]. In this work, to further demonstrate the reliability of our results, we have developed in vitro biofilms of a *P. aeruginosa* strain producing the Green Fluorescent Protein (GFP), and exposed them to stress conditions, mimicking those found in the CF lung environment between antibiotic treatments, i.e., starvation and low drug concentrations. The development/maintenance of culturable persisters and VBNC cells was also tested by a novel flow cytometry protocol and the behavior of tobramycin, ciprofloxacin and six further antibiotics and/or their combinations was evaluated.

## 2. Results

### 2.1. Construction and Analysis of the GFP-Producing Strain P. aeruginosa PAEG1

The strain *P. aeruginosa* PAEG1 was obtained by transforming *P. aeruginosa* PAO1-N with the recombinant shuttle vector pHERD30T encoding the modified thermostable GFP(ASV) (Figure 1).

As the *gfp(ASV)* gene was cloned under the influence of the araBAD promoter, the ability of the recombinant strain to produce the GFP protein was assessed by fluorescence microscopy and flow cytometry, after the exposure for 3 h to 0.2% arabinose.

The microscopy assays evidenced an increasing fluorescence emission by *P. aeruginosa* cells over time, as shown in Figure 2.

The observed data were then confirmed in flow cytometry assays, which reported the same increase of fluorescence intensity in the 3 h-experiment, as evidenced in Figure 3.

### 2.2. P. aeruginosa PAEG1 Quantification by GFP-Flow Cytometry

The detection of the GFP(ASV) protein was used to quantify *P. aeruginosa* by flow cytometry in planktonic and sessile cultures of *P. aeruginosa* PAEG1, as shown in Table 1.

The first set of experiments was performed on exponential phase *P. aeruginosa* cultures and flow cytometry viable counts were compared to plate counts; although causing an increase of GFP production, an arabinose concentration of 0.2% was not satisfying to obtain a reliable and reproducible count. The increase of the arabinose concentration to 0.5% allowed us to reach counts of fluorescent cells corresponding up to 37.45% of the culturable population. Flow cytometry counts of gentamicin (20× MIC) exposed *P. aeruginosa* biofilms were then compared with those obtained by both culture and qPCR, showing the ability of the former technique to detect up to 37.15% of the total bacterial population. The results obtained by flow cytometry were thus considered to be 30% of the real bacterial counts and the obtained results were corrected accordingly throughout the work.

### 2.3. P. aeruginosa PAEG1 Antibiotic Susceptibility and Piperacillin Persistence Assays

The Minimal Inhibitory Concentration (MIC) of nine anti-pseudomonas drugs against the engineered strain *P. aeruginosa* PAEG1 was evaluated (Table 2).

Piperacillin at 1000× MIC was used to select biofilm persistent cells after 24 h exposure and 1/4× MIC of each of the additional antibiotics was used in monitoring assays over time of *P. aeruginosa* biofilms exposed to stress conditions.

*P. aeruginosa* PAEG1 biofilms were developed in Lysogenic Broth (LB) for 24 h; then, the medium was replaced with LB broth containing 4 mg/mL (1000× MIC) piperacillin for 24 h. Immediately before and after antibiotic exposure, the amounts of culturable, Total Viable Cells (TVCs) and VBNC cells were determined, as shown in Figure 4.

Also, in these assays, GFP-based flow cytometry counts resulted about 30% of the qPCR-counted cells. The high dose of the antibiotic mostly affected the culturable *P. aeruginosa* population, which significantly (*p* < 0.05) decreased from 7.59 × 10^6^ to 2.91 × 10^5^ CFU/mL (almost 1.5 log), while the TVCs showed only a slight (3.08 × 10^7^ vs. 2.38 × 10^7^ cells/mL) decrease and the VBNC cells amount was unchanged.

### 2.4. P. aeruginosa Biofilm Exposure to Stress Factors and VBNC Cells Development

After exposure to 1000× MIC piperacillin (T0), biofilms were washed with phosphate buffered saline (PBS) and subcultured in Non-Nutrient (NN) broth, alone or supplemented with 1/4× MIC of each tested antibiotic. Biofilms were maintained in these conditions for 45 days, assessing the abundance of CFUs and TVCs cells every 15 days (T15, T30 and T45) (Figure 5).

TVCs exhibited a major decrease in the first 15 days of stress exposure, then they remained generally stable (reporting the typical plateau characterizing the persistent cells killing curves) until the end of the experiment in almost all tested conditions, except for colistin- or ciprofloxacin-exposed biofilms, where they continuously decreased in the first 30 days and then showed a sudden increase at the last time point.

The amount of the *P. aeruginosa* culturable population was the most variable between stress conditions, exhibiting a drug-specific pattern. In NN-, tobramycin- and fosfomycin-exposed biofilms, CFUs abundance showed an increase in the first 15 days, reaching a concentration of about 1–2 × 10^6^ CFU/mL, and remained stable for the following 30 days. When exposed to colistin, ceftazidime and ceftolozane/tazobactam they resulted overall unchanged until the end of the experiment, while, upon exposure to meropenem, their amount uniformly decreased, reaching the final concentration of 2.35 × 10^4^ CFU/mL, the lowest detected among all experimental conditions; on the contrary, ceftazidime/avibactam-exposed biofilms reported an increase of the culturable population, which finally matched the TVC amount (about 5 × 10^6^ cells/mL) after 45 days. Finally, ciprofloxacin subinhibitory concentrations determined a great decrease of CFUs in the first month of antibiotic exposure, reaching 4.63 × 10^4^ CFU/mL, followed by a sudden increase up to 1.20 × 10^6^ CFU/mL, at T45, where no discrepancy with TVC amount was evidenced.

The selection of ciprofloxacin- and ceftazidime/avibactam resistant mutants was excluded by the MIC determination of several isolates grown on Cystine-Lactose-Electrolyte-Deficient (CLED) agar plates at the last time point, which resulted unchanged compared with that determined before starting the experiment (Table 2).

The VBNC portion of the total *P. aeruginosa* population was then calculated as the difference between TVCs and CFUs; only values ≥0.5 log were considered (Figure 6).

In *P. aeruginosa* biofilms not exposed to any antibiotic, CFUs and TVCs counts always matched, highlighting the absence of VBNC cells in biofilms only starved. In tobramycin-exposed samples, the same result was obtained at T15; subsequently, a stable amount of VBNC cells (i.e., about 1 × 10^6^ cells/mL) was recovered until the end of the experiment. Subinhibitory concentrations of both ciprofloxacin and colistin resulted in a decrease of VBNC cells for the first 30 days of exposure; compared to the other antibiotics, ciprofloxacin-exposure resulted in a significantly (*p* < 0.001) lower amount of VBNC *P. aeruginosa*, and eventually in their loss. On the contrary, colistin-exposure was associated to a subsequent increase of non-culturable cells, which at the end were 2.81 × 10^6^ cells/mL, an amount similar to that evidenced at the first time point (T15). In the presence of fosfomycin, meropenem, ceftazidime and ceftolozane/tazobactam, VBNC cells remained overall unchanged throughout the 45 days of antibiotic exposure, i.e., 2–4 × 10^6^ cells/mL. Meropenem-exposed biofilms showed the highest proportion (98.4%) of VBNC cells inside the TVCs counted at the end of the experiment. Finally, it is to note that in the presence of the association ceftazidime/avibactam, while VBNC cells were detected at T15 and T30, at T45 they disappeared, suggesting a role for the β-lactamases inhibitor avibactam in contrasting *P. aeruginosa* VBNC cells maintenance over time.

### 2.5. Biofilm and c-di-GMP Production under Exposure to Sub-MIC of Antibiotics

Finally, we measured the influence of sub-MIC of the eight different antibiotics/antibiotic combinations on the amount of biofilm matrix and the production of the second messenger c-di-GMP. To do this, *P. aeruginosa* biofilms were developed in Mueller Hinton (MH) broth alone or supplemented with 1/4× MIC of each antibiotic and the amount of biofilm (detected by crystal violet staining) and c-di-GMP (detected by the fluorescence emitted by *P. aeruginosa* PAO1 strain, producing GFP in response to c-di-GMP levels) produced in the absence/presence of the antibiotics were compared. The results are reported in Figure 7.

Biofilm development was significantly (*p* < 0.05) enhanced in the presence of subinhibitory concentrations of ciprofloxacin, tobramycin, ceftazidime/avibactam and ceftolozane/tazobactam; on the contrary, it was significantly (*p* < 0.05) repressed upon exposure to colistin, fosfomycin, meropenem and ceftazidime. The c-di-GMP production was increased only in presence of tobramycin and ciprofloxacin, while it was repressed when biofilms were grown with colistin or β-lactam antibiotics.

## 3. Discussion

*P. aeruginosa* persistent cells, including the VBNC forms, are involved in the chronic lung infection recurrence in CF patients, being able to tolerate antibiotic treatment and to prevent infection eradication [5,15].

The effectiveness of the antibiotic treatment is routinely evaluated by culture-based microbiological assays, which, however, fail to detect these bacterial forms. Intermittent and chronic lung infections are treated with repeated cycles of antibiotic treatment, which usually results in a temporary symptoms remission. Between the therapeutic cycles, low (subinhibitory) antibiotic concentrations can still be found in the lung; the same thing can also happen even during antibiotic treatments in the deepest layers of microbial biofilms [16], where bacterial cells are exposed to a wide range of additional stress, including starvation [17], perhaps the most important persisters-selecting factor [7].

Our group has recently described the involvement of sub-MIC ciprofloxacin and tobramycin (two drugs usually adopted in the treatment of *P. aeruginosa* CF lung infections [18]) in the persistence of *P. aeruginosa* VBNC cells in an in vitro biofilm model, highlighting a major contribution of tobramycin in their long time (almost six months) maintenance [11].

In this paper, we used the previously adopted experimental settings with some improvements. First of all, to better detect *P. aeruginosa* live cells by flow cytometry, the GFP-producing *P. aeruginosa* strain PAEG1 was constructed and used to develop the biofilm models. GFP production is a reliable marker of viability that allowed us to overcome the weaknesses inherent to flow cytometry counts after live/dead staining. As a matter of fact, the two dyes (SYBR Green and propidium iodide) used could bind biofilm intrinsic and non-specific targets, like extracellular DNA [19], thus affecting the live bacterial quantification; moreover, the distinction between live (SYBR Green stained) and dead (propidium iodide stained) cells can be affected by the variable cell membrane permeability, which characterize stress-exposed, but still alive, cells [20]. The *P. aeruginosa* PAEG1 strain resulted a suitable study model, since the fluorescent protein was adequately expressed by the host strain and easily detected in both microscopy and flow cytometry assays. Although only about 30% of the total live *P. aeruginosa* population emitted a detectable fluorescence, the same rate of fluorescent cells was recovered in both planktonic and sessile cultures, suggesting a systematic error, most likely due to the flow cytometer itself. One possible explanation is that the modified GFP(ASV) showed excitation and emission wavelengths of 500 and 510 nm, respectively, while the flow cytometer laser was set by default on 488 nm for excitation and 525/30 nm for emission; such discrepancy could account for the observed low amount of GFP-expressing bacterial cells, likely due to the loss of the cells emitting a low fluorescent signal, which resulted however unaffected by the sample type, i.e., planktonic or biofilm-embedded.

As the second modification of the previously performed assays on VBNC induction and maintenance in *P. aeruginosa* biofilms, in the present work, we added a pre-treatment of *P. aeruginosa* biofilms with 1000× MIC to select the persistent subpopulation already present before the subculture in nutrient-depleted medium without/with sub-MIC of antibiotics. This step was added since the presence of persisters most likely mirrors what happens in the lung environment after the antibiotic treatment, i.e., with the survival of persistent *P. aeruginosa* cells, both culturable and VBNC [14,21]. Although we are aware that the piperacillin concentration (1000× MIC) we used cannot be reached in vivo, the aim of this assay was to select true persisters, avoiding the recovery of adaptive resistant mutants [22]. The obtained results documented the shift of most *P. aeruginosa* population cells from the culturable to the VBNC state after high doses of antibiotic exposure, in agreement with what reported by Deschaght [5] in CF patients undergoing antibiotic therapy with tobramycin, meropenem, colimycine, piperacillin/tazobactam or ceftazidime.

The selected persistent cells, both culturable and non-culturable, were then maintained for 45 days in absence of nutrients and presence of 1/4× MIC of different antibiotics and the development of both populations was monitored over time, with a special focus on the proportion of VBNC cells, with respect to the specific drugs.

A remarkable result was the total loss of non-culturable cells in absence of antibiotics; such outcome likely derived from the observed TVCs reduction and CFUs increase (Figure 5) observed at T45; it represents the only condition giving no VBNC cells starting from the first time point (15 days) from piperacillin-persisters selection. The observed behavior could be explained by: (i) the proliferation of piperacillin-persistent cells after the removal of the antibiotic; (ii) a programmed cell death of a *P. aeruginosa* subpopulation, which allowed the survival of a live and culturable subpopulation [23], without development of the VBNC phenotype.

The observed involvement of ciprofloxacin and tobramycin in *P. aeruginosa* VBNC maintenance confirmed our previous results obtained with six months-monitored starvation- and antibiotic-exposed biofilms [11]. Ciprofloxacin-exposure, after a greater amount of non-culturable cells in the first 15 days, resulted in a total lack of VBNC cells at the end of the experiment (45 days of sub-MIC antibiotic-exposure), possibly replaced by a new proliferating subpopulation; a behavior similar to what was previously observed after four months of monitoring [11]. No increase of the ciprofloxacin MIC was observed against *P. aeruginosa* isolates resulted culturable at T45, thus excluding the induction of a resistant, predominant subpopulation [24]. The observed increase of culturable cells could be attributed to cell proliferation thanks to the programmed death of the VBNC subpopulation.

Tobramycin induced the development and maintenance of a *P. aeruginosa* VBNC subpopulation after one month of exposure till the end of the experiment, in line with what was previously observed [11] and confirming the pivotal role of protein synthesis inhibitors in the VBNC cells induction [25].

Cell wall inhibiting antibiotics and colistin subinhibitory concentrations were also tested. As far as concern colistin, it caused a constant reduction over time of the VBNC subpopulation, while the culturable ones were unaffected by the presence of the drug; however, at the last time point, the appearance of a VBNC subpopulation was observed. Colistin is well known to act on dormant cells [26], in agreement with the observed reduction of the VBNC cells over time; however, their late induction is surprising and is currently being evaluated in further biofilm models by monitoring the expression of specific target genes, such as *phoP* and *pmrA* [27].

The *P. aeruginosa* response to different cell wall inhibiting antibiotics was not uniform: fosfomycin resulted as the least efficient in reducing the TVCs over time and induced the highest amount of VBNC cells at the end of the experiment, compared to all tested antibiotics; exposure to meropenem induced a uniform decrease of CFUs and increase of the VBNC proportion among the total viable *P. aeruginosa* population, which resulted in the highest at T45 (Figure 5). Regarding cephalosporines and their combinations with β-lactamases inhibitors, only the presence of avibactam lead to the loss of the VBNC subpopulation, while ceftazidime alone and ceftolozane/tazobactam maintained a stable set of non-culturable cells throughout the experiment. The role of the bacterial cell wall and its modifications in the VBNC state are well-known [28], thus an involvement of antibiotics affecting cell wall synthesis in its induction is not surprising, as previously observed by Pasquaroli and colleagues [10]. However, since the antibiotics were used at subinhibitory concentrations, their influence on the VBNC state induction is likely due to gene expression regulation rather than growth inhibition [12].

The combination ceftazidime/avibactam lead to a continuous increase of CFUs over time, until the loss of the non-culturable subpopulation; also in this case, the development of resistant mutants was excluded. Considering the results obtained with ceftolozane/tazobactam, the most likely explanation for the observed discrepancy in VBNC cells maintenance is to be attributed to the different chemical nature of the β-lactamase inhibitor (non β-lactam vs. β-lactam for avibactam and tazobactam, respectively) and the subsequent elicited bacterial response via specific receptors. Further experiments are currently ongoing to verify this hypothesis.

In an attempt to determine the putative effectors of *P. aeruginosa* VBNC cells induction, biofilm and c-di-GMP production were measured upon exposure to sub-MIC antibiotic concentrations. Biofilm production was enhanced in the presence of sub-MIC of ciprofloxacin, tobramycin, avibactam and tazobactam and it was repressed in the presence of sub-MIC of colistin and all tested cell wall targeting drugs. Ciprofloxacin, tobramycin and β-lactams effects are in agreement with previous observations [29,30,31]. For the former two antibiotics, the involvement of the protein Arr, responsible for the synthesis/degradation of c-di-GMP, has been proposed as the main positive regulatory pathway [29]. Accordingly, even the second messenger amount was higher when biofilms were developed in presence of ciprofloxacin and tobramycin, compared to antibiotic-free medium. The other antibiotics, in particular the β-lactams, generally determined a reduction of c-di-GMP levels, except for fosfomycin, which did not influence its production. The β-lactam antibiotics, specifically ceftazidime [31], have been described to reduce biofilm formation by repressing the production of different components of the *P. aeruginosa* biofilm matrix and bacterial motility, which is pivotal in the first stages of biofilm formation, in agreement with our results. The exposure to β-lactamases inhibitors was shown to reduce biofilm production as well, in particular the combination piperacillin/tazobactam [32], while our data demonstrated enhanced ability to shift to the sessile state in presence of both avibactam and tazobactam. Whether this discrepancy is actually due to the antibiotic (penicillin or cephalosporine) of the combination is currently under evaluation, together with their effect on the *P. aeruginosa* VBNC subpopulation.

When considering together the induction of VBNC cells, biofilm production and c-di-GMP levels, our results succeeded in demonstrating a direct correlation among these three bacterial responses only for tobramycin-exposed biofilms; for all other antibiotics VBNC cells induction related neither to biofilm production or to c-di-GMP expression.

Overall, the obtained data showed different effects exerted by subinhibitory antibiotic concentrations on *P. aeruginosa* biofilms. Two of the main goals of antimicrobial research are the infectious biofilms eradication and to avoid the insurgence of antimicrobial resistance. However, the present work shows that each antibiotic can exert a specific effect on the bacterial viability and gene expression, not following a unique and defined pathway. *P. aeruginosa* is known to possess several redundant regulatory mechanisms [33,34], based on different receptors and regulatory proteins, which could contribute to the observed drug-specific response in order to overcome stress conditions and to ensure bacterial survival.

## 4. Materials and Methods

### 4.1. Bacterial Strains, Plasmids, Media, Antibiotics and Chemical Compounds

The laboratory strains *P. aeruginosa* PAO1-N, its isogenic recombinant version carrying the pcdrA plasmid, encoding the GFP, and the Escherichia-Pseudomonas shuttle vector pHERD30T were kindly provided by Professor Paul Williams (Center for Biomolecular Sciences, University of Nottingham, Nottingham, UK). The Pfu polymerase and the plasmid pGFP29, containing a mutated *gfp* gene copy, encoding a thermostable GFP protein (GFP(ASV)), containing an additional tail of 13 amino acids (RPAANDENYAASV) at the *C*-terminal, were kindly provided by Roberto Spurio (School of Biosciences and Veterinary Medicine, University of Camerino, Camerino, IT). The strain *P. aeruginosa* ATCC 27853 belongs to the strains collection of Microbiology section of Department of Life and Environmental Sciences, Polytechnic University of Marche, Ancona, IT.

All bacteriological media were purchased from Oxoid (Thermo Fisher Scientific, Waltham, MA, USA), while antibiotics and arabinose from Sigma-Aldrich (Saint Luis, Missouri, USA).

Bacterial strains were grown on LB Broth, subcultured on McConkey agar plates and conserved as stock cultures in LB broth supplemented with 20% glycerol.

### 4.2. GFP-Producing P. aeruginosa Recombinant Strain Construction

To produce a *P. aeruginosa* PAO1-N recombinant strain, able to produce a stable GFP, the *gfp(ASV)* gene was cloned into the pHERD30T vector using the In-Fusion HD Cloning Plus kit (Takara Bio USA, Inc., Mountain View, CA, USA), according to the manufacturer instructions. Briefly the gene was amplified from the pGFP29 plasmid by PCR assays using 0.1 ng/µL of plasmid, 5 U of Pfu polymerase and 0.5 µM of the primer pair GFP-F: 5′—GAGATATACATACCCATGCGTAAAGGAGAAGAACTTTTCAC—3′/ GFP-R: 5′—ATTCTTATCAGATCCGGAGCTGCATGTGTCAGAGGTTTTC—3′. The primers were designed to present 15 bp (underlined) complementary to the multiple cloning site of the pHERD30T vector. Reactions were performed using an annealing temperature of 45 °C and the amplification product (898 bp) was visualized by gel electrophoresis in 1.5% agarose gel in 1× TAE buffer. The pHERD30T vector was linearized by digestion with NcoI restriction enzyme (Promega, Madison, WI, USA), then 100 ng of both vector and PCR amplicon were used in the In-Fusion cloning reaction, performed at 50 °C for 15 min and maintained in ice. The recombinant vector, visualized by gel electrophoresis, was propagated in the strain *E. coli* stellar (provided with the kit), then extracted using the GenElute™ Plasmid Miniprep Kit (Sigma-Aldrich) and introduced in *P. aeruginosa* PAO1-N by electroporation. Reactions were performed using 90 µL of competent *P. aeruginosa* cells and 100 ng of the recombinant plasmid in a Gene PulserXcell ™ electroporator (Bio-Rad, Hercules, CA, USA) with the following setting: 2.5 kV, 25 µFD, 200 OHMs, 5 ms. The transformed cells were diluted in 1 mL of LB broth, incubated for 1–3 h at 37 °C and then plated on LB agar containing 20 µg/mL gentamicin. The growing colonies were amplified and then used in the subsequent assays.

### 4.3. GFP Induction and Detection

The obtained recombinant *P. aeruginosa* PAEG1 strain was assessed for its ability to produce the GFP in fluorescence microscopy and flow cytometry assays. As the *gfp(ASV)* gene was cloned under the control of an arabinose-dependent promoter, its expression was induced by incubating *P. aeruginosa* PAEG1 in LB broth supplemented with 0.2% arabinose at 37 °C for 0, 1, 2 and 3 h.

For fluorescence microscopy assays, 1 mL of induced cultures was filtered in polycarbonate 0.22 µm black filters (Millipore, Burlington, MA, USA) and observed in an Olympus BX51 microscope, using 100× magnification and a blue light filter (450–490 nm).

In flow cytometry assays, 200 µL of induced cultures were centrifuged at 16,000× *g* and resuspended in PBS to be read by the cytometer. Assays were performed in a Guava Millipore cytometer and analyzed by GUAVASOFT 2.2.3 software. Both side scatter and green (GRN) fluorescence were gated to discriminate bacterial cells from the background. The GFP fluorescence was excited at 488 nm, a threshold value was set in the GRN channel and the emissions were read at 525/30 nm. Signal detection was enhanced by logarithmic amplification (4 decades); a total number of 20,000 events/replicate was analyzed to increase statistical significance. The GFP mean fluorescence emission was plotted against the incubation time. Assays were run in technical duplicate and biological triplicate.

### 4.4. Antibiotic Susceptibility Assays

The MIC values of each antibiotic was determined against *P. aeruginosa* using the broth microdilution method according to the CLSI guidelines [35]. When testing the combinations antibiotic/inhibitors, a fixed concentration of 4 µg/mL was adopted for β-lactamases inhibitors. *P. aeruginosa* ATCC 27853 was used as the reference strain.

### 4.5. Biofilm Formation, Persistence Assays and Maintenance in Stress Conditions

Unless otherwise stated, *P. aeruginosa* PAEG1 was always grown and maintained in media supplemented with gentamicin 20 µg/mL, to allow the maintenance of the recombinant pHERD30T vector. *P. aeruginosa* biofilms were grown in LB medium at 37 °C overnight in 140 mm petri dishes, each containing fifteen 24 × 24 mm cover glasses, and challenged with LB broth supplemented with 1000× MIC piperacillin for 24 h at 37 °C. Then the medium was removed, biofilms were washed with PBS and subcultured in Non-Nutrient (NN) broth without/with 1/4× MIC of each antibiotic, as previously described [11]. The samples were maintained in these conditions for 45 days, refreshing the NN medium once a week.

Biofilms were analyzed before and after exposure to piperacillin and every 15 days after subculture in NN broth (T0, T15, T30 and T45). At each time point, 3 cover glasses/condition were removed from the plates and dipped in 2.5 mL of PBS. The biofilm adherent to the glasses was removed mechanically and briefly sonicated.

### 4.6. Biofilm Analysis, P. aeruginosa Populations Identification

The different *P. aeruginosa* populations were quantified in the detached biofilms as follows:The culturable population was counted by plate count assays, performed on CLED agar plates, evaluated after 24, 48 and 72 h from the inoculum.The TVCs were determined by both *ecfX*-qPCR and flow cytometry assays; the formers were performed using a previously described protocol [14], demonstrated to detect only live cells, while the latter as described above, after 1 h incubation of 1:5 diluted biofilms in LB supplemented with 0.2 or 0.5% arabinose. TVCs were quantified as average between *P. aeruginosa* amounts counted by qPCR and flow cytometry.The VBNC population was estimated as discrepancy between TVCs and culturable cells. The obtained values were not considered when the discrepancy was ≤0.5 log.

### 4.7. Biofilm and c-di-GMP Production under Exposure to Sub-MIC Antibiotics

The role of the sub-inhibitory concentrations of the tested antibiotics was verified even in biofilm and c-di-GMP production, using a *P. aeruginosa* PAO1-N strain, carrying a plasmid encoding the GFP protein under the influence of the *cdrA* promoter and responding to c-di-GMP levels. *P. aeruginosa* biofilms were developed in 96-well flat bottom dark plates in MH broth, with/without 1/4× MIC of each antibiotic. When testing the combinations ceftazidime/avibactam and ceftolozane/tazobactam, the β-lactamase inhibitor concentration was set at 4 µg/mL.

Biofilm production was then quantified as previously described [11]; for the c-di-GMP detection, the GFP fluorescence was measured in *P. aeruginosa* biofilms. After removing the planktonic phase and washing the plate wells with water, biofilms were mechanically detached and resuspended in PBS. The plate was then read using a Synergy HT MicroPlate Reader Spectrophotometer (BioTek, Winooski, VT, USA), measuring both biofilms optical density (OD) at 550 nm and fluorescence (excitation at 485 nm, emission at 535 nm). The fluorescent signal was normalized by calculating the Fluorescence/OD_550_ ratio.

### 4.8. Statistical Analysis

The significance of the amount of *P. aeruginosa* populations (culturable, TVCs and VBNC) in the different stress conditions, as well as the biofilm and c-di-GMP production was assessed by Student’s *t*-test (threshold, 0.05).

## 5. Conclusions

The *P. aeruginosa* persistent and VBNC cells hamper the eradication of chronic CF lung infections and antibiotic treatment seems to be crucially involved in their induction and maintenance in the lung environment. The present work highlighted the specific contribution of different drugs to *P. aeruginosa* persisters and VBNC cells induction, underlining the major role of protein and cell wall synthesis inhibition in their development; on the contrary, DNA replication inhibition and the use of non-β-lactam β-lactamase inhibitors seem to limit the insurgence of the VBNC phenotype, thus constituting promising strategies to counteract *P. aeruginosa* resilience.

Moreover, the variety of the bacterial response to each antibiotic seems to suggest a complex and redundant regulatory system, based on different regulatory proteins, able to mediate *P. aeruginosa* adaptation to each specific stress condition. The characterization of the molecular bases of the antibiotic-induced *P. aeruginosa* gene expression seems thus a pivotal approach to shed light on the factors underpinning the failure of *P. aeruginosa* lung infection eradication and to design novel, more effective therapeutic treatments.

## Figures and Tables

**Figure 1 ijms-22-01628-f001:**
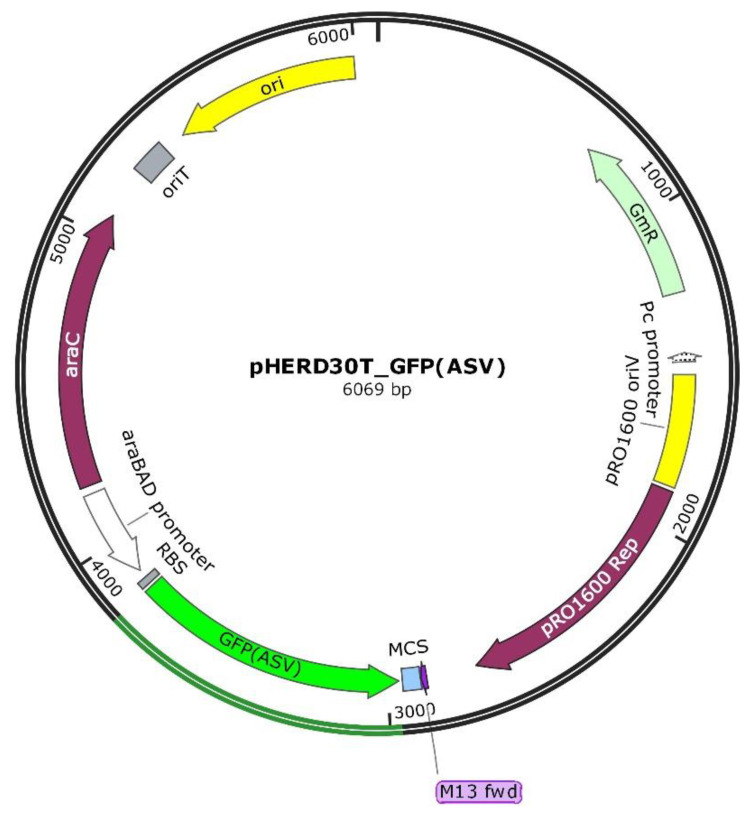
The recombinant pHERD30T_GFP(ASV) vector. The recombinant pHERD30T vector, encoding the GFP(ASV) protein was obtained by cloning the *gfp(ASV)* gene into the original vector using the In-Fusion HD Cloning Plus kit; the vector map was designed with the software SNAPGENE v. 4.2.6.

**Figure 2 ijms-22-01628-f002:**
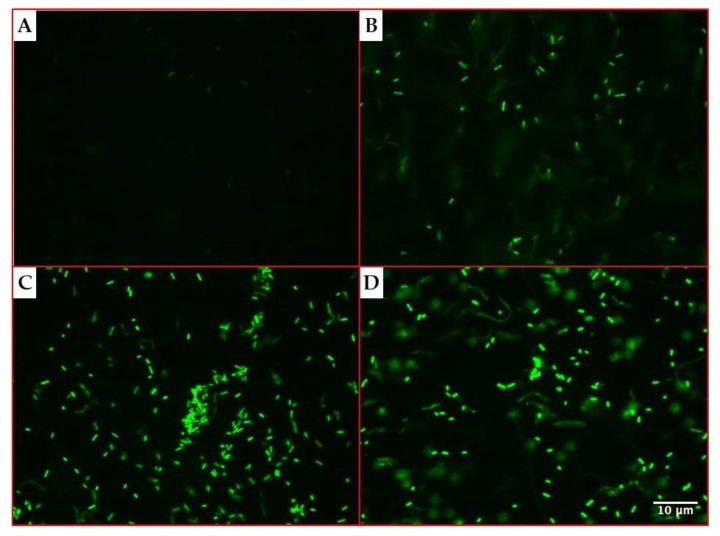
Green fluorescent protein production by *P. aeruginosa* PAEG1 detected by fluorescence microscopy. The production of the GFP(ASV) was assessed in a culture of the recombinant strain *P. aeruginosa* PAEG1 in LB broth supplemented with 0.2% arabinose, incubated for 0 (**A**), 1 (**B**), 2 (**C**) and 3 h (**D**) at 37 °C. Aliquots of the culture were filtered in polycarbonate black filters (0.22 µm diameter) and observed at 100× magnification using a blue light laser.

**Figure 3 ijms-22-01628-f003:**
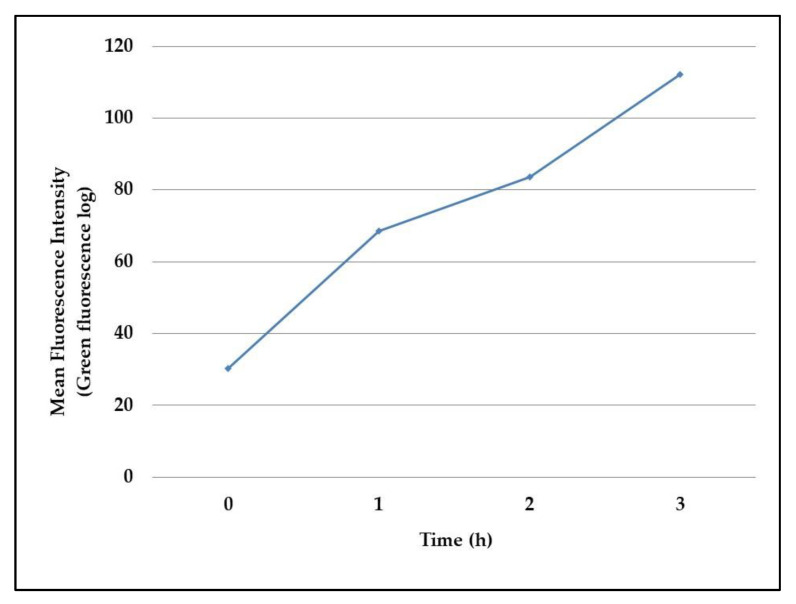
Green fluorescent protein production by *P. aeruginosa* PAEG1 detected by flow cytometry assays. The GFP(ASV) fluorescence was measured by flow cytometry in a culture of *P. aeruginosa* PAEG1 in LB broth containing 0.2% arabinose, incubated for 0, 1, 2 and 3 h at 37 °C. Every hour, a 200 µL aliquot of the culture was withdrawn, washed in phosphate buffered saline (PBS) and measured in the green channel. About 20,000 events/samples were considered, and the results are expressed as mean fluorescence intensity for each sample.

**Figure 4 ijms-22-01628-f004:**
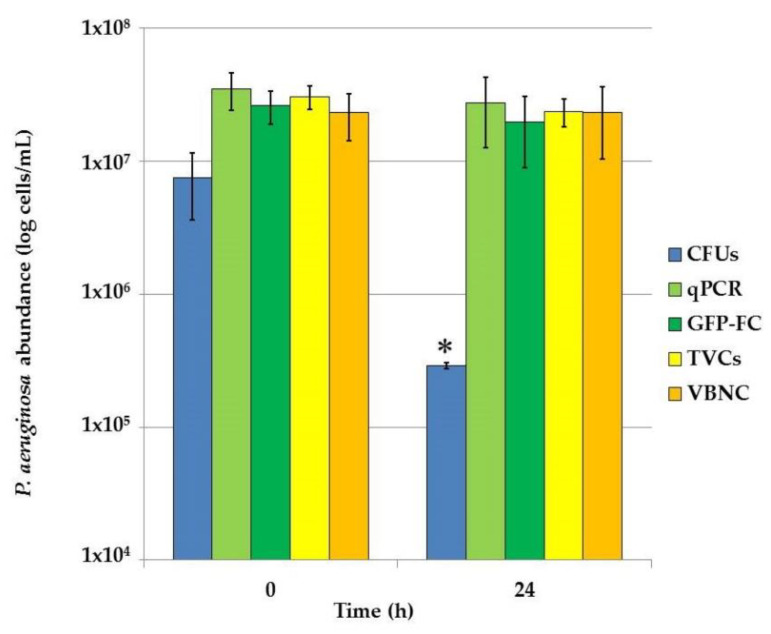
Piperacillin persistence assays of *P. aeruginosa* biofilms. *P. aeruginosa* PAEG1 biofilms were developed in LB broth for 24 h at 37 °C; then, the medium was removed and replaced with LB broth containing piperacillin 4 mg/mL (1000× MIC) and biofilms were maintained in these conditions for 24 h at 37 °C. Immediately before (0 h) and after (24 h) antibiotic exposure, the amounts of culturable persisters (CFUs) and Total Viable Cells (TVCs) were determined by plate count and qPCR/ GFP-based flow cytometry (GFP-FC) assays, respectively. The amount of viable but non-culturable (VBNC) cells were determined as the difference between TVCs and CFUs. The results are expressed as average of three biological replicates ± standard deviation. *, *p* < 0.05.

**Figure 5 ijms-22-01628-f005:**
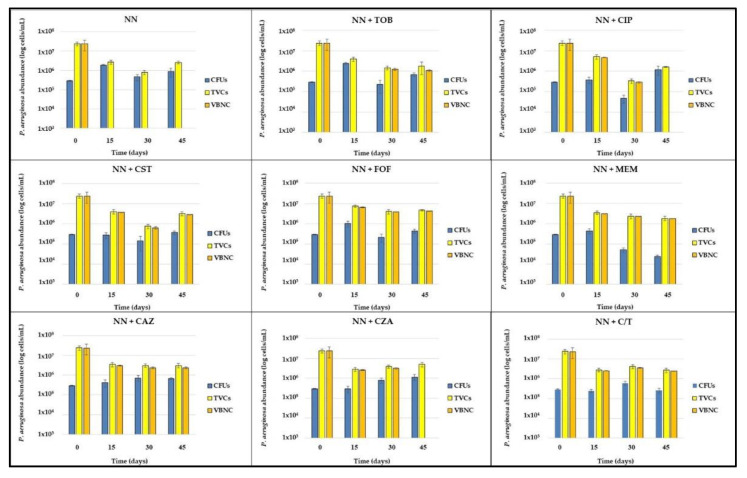
*P. aeruginosa* populations monitoring in starved and antibiotic-exposed biofilms. *P. aeruginosa* biofilms, developed in rich medium, were exposed 1000× MIC piperacillin for 24 h (T0), then subcultured in non-nutrient broth, alone or supplemented with 1/4× MIC of different antibiotics, and maintained in these conditions for 45 days. The amount of the different *P. aeruginosa* subpopulations was monitored every 15 days (T15, T30 and T45) by a combination of cultural, qPCR and flow cytometry assays. The results are expressed as average of three biological replicates ± standard deviation. NN, non-nutrient; TOB, tobramycin; CIP, ciprofloxacin; CST, colistin; FOF, Fosfomycin; MEM, meropenem; CAZ, ceftazidime; CZA, ceftazidime/avibactam; C/T, ceftolozane/tazobactam; CFUs, colony forming units; TVCs, total viable cells; VBNC, viable but non-culturable cells.

**Figure 6 ijms-22-01628-f006:**
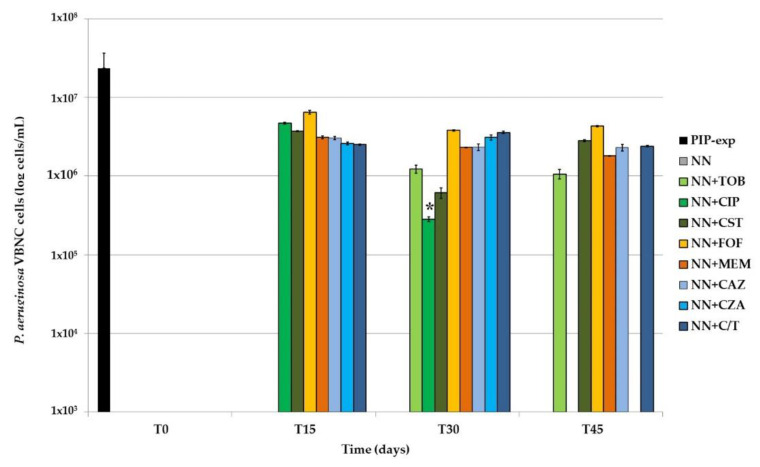
VBNC cells amount in *P. aeruginosa* biofilms exposed for 45 days to starvation and 1/4× MIC of eight different antibiotics/antibiotic combinations. After exposure to 1000× MIC piperacillin (PIP-exp, T0), *P. aeruginosa* biofilms were subcultured in Non-Nutrient (NN) broth, alone or supplemented with 1/4× MIC of tobramycin (TOB), ciprofloxacin (CIP), colistin (CST), fosfomycin (FOF), meropenem (MEM), ceftazidime (CAZ), ceftazidime/avibactam (CZA) or ceftolozane/tazobactam (C/T). The samples were maintained in these conditions for 45 days and assessed for the amount of VBNC cells every 15 days (T15, T30 and T45). The results are reported as average of three biological replicates ± standard deviation. *, *p* < 0.001.

**Figure 7 ijms-22-01628-f007:**
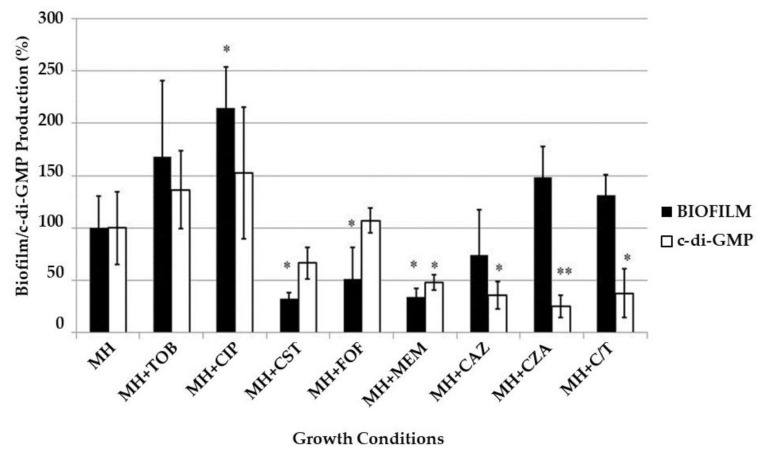
*P. aeruginosa* PAO1 biofilm and c-di-GMP production after exposure to sub-MIC of eight different antibiotic/antibiotic combinations. *P. aeruginosa* biofilms were developed for 24 h at 37 °C in Mueller-Hinton (MH) broth alone or supplemented with1/4× MIC of tobramycin (TOB), ciprofloxacin (CIP), colistin (CST), fosfomycin (FOF), meropenem (MEM), ceftazidime (CAZ), ceftazidime/avibactam (CZA) or ceftolozane/tazobactam (C/T). Biofilm and c-di-GMP production were measured by crystal violet staining and by GFP fluorescence measurement, respectively. The obtained values were normalized by biofilms OD_625_ and are expressed as percent of the value obtained with the biofilms developed without antibiotic; the results are reported as average of three biological replicates ± standard deviation. *, *p* < 0.05; **, *p* < 0.001.

**Table 1 ijms-22-01628-t001:** GFP-based flow cytometry (GFP-FC) viable counts of *P. aeruginosa* PAEG1 grown in broth or in biofilms.

Sample	Plate Count (CFU/mL)	qPCR Count (Cells/mL)	GFP-FC Counts (Cells/mL)
Broth culture (OD_600_ = 0.1) 0.2% arabinose induction	2.54 × 10^7^ (±0.516)	/	4.03 × 10^5^ (±0.001)
Broth culture (OD_600_ = 0.1) 0.5% arabinose induction	2.12 × 10^7^ (±0.475)	/	7.94 × 10^6^ (±1.850)
Gentamicin-exposed biofilm 0.5% arabinose induction	1.88 × 10^7^ (±0.679)	1.65 × 10^8^ (±0.075)	6.13 × 10^7^ (±0.827)

Counts are reported as average of three biological replicates (± standard deviation); /= not performed.

**Table 2 ijms-22-01628-t002:** Susceptibility (MIC values) of *P. aeruginosa* PAEG1 against nine antibiotics/antibiotic combinations.

Antibiotic	MIC (µg/mL)
Ciprofloxacin	0.25
Tobramycin	1
Colistin	4
Fosfomicin	32
Piperacillin	4
Meropenem	4
Ceftazidime	1
Ceftazidime/avibactam	1/4
Ceftolozane/tazobactam	0.5/4

## Data Availability

Not applicable.

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
