# Peer review of "Contribution of Drugs Interfering with Protein and Cell Wall Synthesis to the Persistence of Pseudomonas aeruginosa Biofilms: An In Vitro Model†"

_ijms, 2021, doi:10.3390/ijms22041628_

Round 1

Reviewer 1 Report

The study is well planned. I would suggest the authors to include the figure S1 in the manuscript for better understanding and comparison. 

In continuation of their previous research on the involvement of sub-MIC ciprofloxacin and tobramycin in the persistence of P. aeruginosa VBNC cells in an in vitro biofilm model, the authors have made an improvement to better detect P. aeruginosa live cells by flow cytometry using the GFP-producing P. aeruginosa strain PAEG1, and to induce and maintain the production of P. aeruginosa biofilms. In this in vitro study, they have evaluated the role of subinhibitory concentrations of different antipseudomonas antibiotics in the maintenance of persistent P. aeruginosa cells in biofilms. Positive correlation between antibiotic exposure, biofilm production and c-di-GMP levels was only observed with tobramycin.

The study is limited to an in-vitro model. The findings are interesting. However, in-vivo study is the main limitation in this study.

Reviewer 2 Report

While the manuscript is well written, with minor linguistic errors, the presented results are an incremental step forward for the authors and some significant assumptions in methods are made that undermine the results. Overall, this manuscript further validates and supports the authors' central hypothesis that subinhibitory use of antibiotics over time increases the development/persistence of VBNC P. aeruginosa; however, the method of calculating VBNC is of major concern.

Major issue:

  • In Table 1, no CFU count is presented for the biofilm culture. Given the order of magnitude difference between CFU and GFP-FC count in planktonic cultures (and not in the direction of GFP-FC detection of viable, but non-culturable cells) indicates that the plasmid may not be well retained or taken up by the viable bacteria, thus skewing the calculation of TVC and VBNC. qPCR count is not an appropriate detection of viability, as it does not indicate viable cells, just presence of nucleic acid signal. While gentamicin was utilized in initial transformation of PAO1 and some subsequent cultures, it was unclear whether the selective pressure for plasmid retention was maintained throughout the experiments.

Minor issue:

  • Clarify in results and methods that method of detection of c-di-GMP. Was this a dual plasmid transfection? Both GFP-FC and PAO1-N wirh GFP under cdrA promoter? Obviously that doesn't make sense, so how does the detection of c-di-GMP related to the central model of VBNC P. aeruginosa presented in this manuscript since different transformed versions of PAO1 are utilized?

Overall, the major issue outlined above undermines the central findings of the manuscript and no major impactful findings or truly novel methods are presented to outweigh these issues.

Round 2

Reviewer 2 Report

The revised manuscript and author responses to my concerns addressed the issues and is much improved. Clarification of specific issues, especially methodology and addition of data that substantiated the conclusions significantly improved the overall merit of this manuscript.

Thank you to the authors for their thoughtful responses and clearly improved manuscript.